# Improving image generative models with human interactions

## Abstract

GANs provide a framework for training generative models which mimic a data distribution. However, in many cases we wish to train a generative model to optimize some auxiliary objective function within the data it generates, such as making more aesthetically pleasing images. In some cases, these objective functions are difficult to evaluate, e.g. they may require human interaction. Here, we develop a system for efficiently training a GAN to increase a generic rate of positive user interactions, for example aesthetic ratings. To do this, we build a model of human behavior in the targeted domain from a relatively small set of interactions, and then use this behavioral model as an auxiliary loss function to improve the generative model. As a proof of concept, we demonstrate that this system is successful at improving positive interaction rates simulated from a variety of objectives, and characterize some factors that affect its performance.

## 1 Introduction

Generative image models have improved rapidly in the past few years, in part because of the success of Generative Adversarial Networks, or GANs (Goodfellow et al., 2014). GANs attempt to train a "generator" to create images which mimic real images, by training it to fool an adversarial "discriminator," which attempts to discern whether images are real or fake. This is one solution to the difficult problem of learning when we don't know how to write down an objective function for image quality: take an empirical distribution of "good" images, and try to match it.

Often, we want to impose additional constraints on our goal distribution besides simply matching empirical data. If we can write down an objective which reflects our goals (even approximately), we can often simply incorporate this into the loss function to achieve our goals. For example, when trying to generate art, we would like our network to be creative and innovative rather than just imitating previous styles, and including a penalty in the loss for producing recognized styles appears to make GANs more creative (Elgammal et al., 2017). Conditioning on image content class, training the discriminator to classify image content as well as making real/fake judgements, and including a loss term for fooling the discriminator on class both allows for targeted image generation and improves overall performance (Odena et al., 2016).

However, sometimes it is not easy to write an explicit objective that reflects our goals. Often the only effective way to evaluate machine learning systems on complex tasks is by asking humans to determine the quality of their results (Christiano et al., 2017, e.g.) or by actually trying them out in the real world. Can we incorporate this kind of feedback to efficiently guide a generative model toward producing better results? Can we do so without a prohibitively expensive and slow amount of data collection? In this paper, we tackle a specific problem of this kind: generating images that cause more positive user interactions. We imagine interactions are measured by a generic Positive Interaction Rate (PIR), which could come from a wide variety of sources.

For example, users might be asked to rate how aesthetically pleasing an image is from 1 to 5 stars. The PIR could be computed as a weighted sum of how frequently different ratings were chosen. Alternatively, these images could be used in the background of web pages. We can assess user interactions with a webpage in a variety of ways (time on page, clicks, shares, etc.), and summarize these interactions as the PIR. In both of these tasks, we don't know exactly what features will affect the PIR, and we certainly don't know how to explicitly compute the PIR for an image. However, we can empirically determine the quality of an image by actually showing it to users, and in this

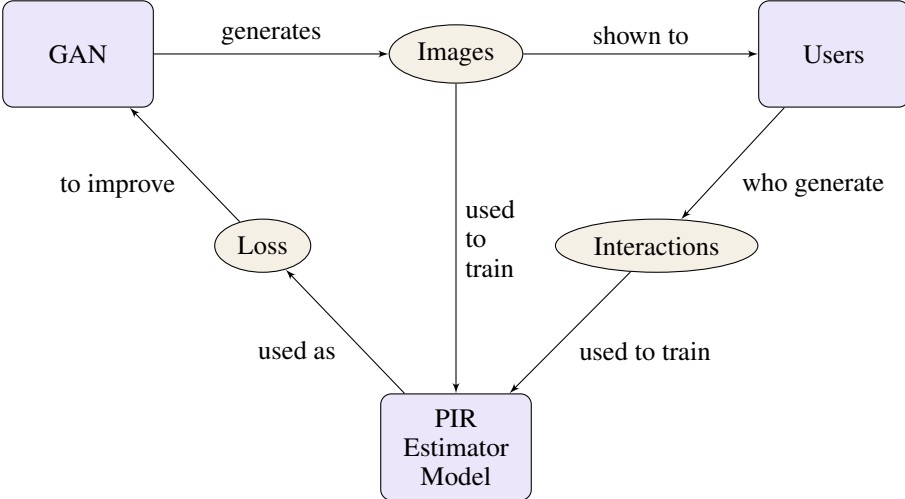

Figure 1: Diagram of our system

paper we show how to use a small amount of this data (results on 1000 images) to efficiently tune a generative model to produce images which increase PIR. In this work we focus on simulated PIR values as a proof of concept, but in future work we will investigate PIR values from real interactions.

## 2 APPROACH

The most straight-forward way to improve an image GAN might be to evaluate the images the model produces with real users at each training step. However, this process is far too slow. Instead, we want to be able to collect a batch of PIR data on a batch of images, and then use this batch of data to improve the generative model for many gradient steps; we want to do this despite the fact that the images the generator is producing may evolve to be very different from the original images we collected PIR data on. In order to do this, we use the batch of image and PIR data to train a "PIR Estimator Model" which predicts PIRs on images. We then use these estimated PIRs at each step as a loss.

Our approach is inspired by the work of Christiano and colleagues (Christiano et al., 2017), who integrated human preference ratings between action sequences into training of a reinforcement learning model by using the preference data to estimate a reward function. However, our problem and approach differ in several key ways. First, we are optimizing a generative image model rather than a RL model. This is more difficult in some ways, since the output space is much higher-dimensional than typical RL problems, which means that scalar feedback (like a PIR) may be harder for the system to learn from. This difficulty is partially offset by the fact that we assume we get "reward" (PIR) information for an image when we evaluate, instead of just getting preferences which we have to map to rewards. Perhaps most importantly, we use our PIR estimation model as a fully-differentiable loss function for training, instead of just using its estimated rewards. This allows us to more effectively exploit its knowledge of the objective function (but risks overfitting).

Our system consists of three components: A generative image model, users who interact with the generated images in some way, and a PIR estimator that models user interactions given an image. See Fig. 1 for a diagram of the system's general operation. The generative model produces images, which are served to users. Using interaction data from these users, we train the PIR estimator model, which predicts PIRs given a background image, and then incorporate this estimated PIR into the loss of the generative model to tune it to produce higher quality images. Below, we discuss each of these components in more detail.

## 2.1 GENERATIVE MODEL

We begin with a GAN[1] (Goodfellow et al., 2014) which we pre-trained to produce images from a target distribution (specifically, landscapes of mountains and coasts). Let $D_{\text{source}}$ be the source estimated by the discriminator, $G$ the generator, and $z$ a noise input to the generator sampled from a multivariate standard normal distribution $\mathcal{N}(0, I)$, and $\mathcal{I}$ be the set of real images shown to the discriminator. Define:

$$L_{\text{fake image}} = E_{z \sim \mathcal{N}(0,I)} \left[ \log P(D_{\text{source}}(G(z)) = \text{fake}) \right]$$
$$L_{\text{fake image fools}} = E_{z \sim \mathcal{N}(0,I)} \left[ \log P(D_{\text{source}}(G(z)) = \text{real}) \right]$$
$$L_{\text{real image}} = E_{i \sim \mathcal{I}} \left[ \log P(D_{\text{source}}(i) = \text{real}) \right]$$

Then the discriminator and generator are trained to maximize the following losses (respectively), where the $w_*$ are weights set as hyperparameters:

$$L_{\text{discriminator}} = L_{\text{fake image}} + L_{\text{real image}}$$
$$L_{\text{Generator}} = L_{\text{fake image fools}}$$

Note that there is a difference between these losses and the standard GAN formulation given in (Goodfellow et al., 2014) – we maximize $L_{\text{fake image fools}} = \log P(\text{classified real})$ rather than minimizing $\log(1 - P(\text{classified real}))$. This seems to result in the generation of slightly better images in practice.

This GAN was trained on a dataset consisting of landscape images of mountains and coastlines (see Appendix C.2 for details of the architecture and training). It is worth noting that this generative model is not photorealistic (see Fig. 3a for some samples). Its expressive capacity is limited, and it has clear output modes with limited intra-mode variability. However, for our purposes this may not matter. Indeed, it is in some ways more interesting if we can tweak this model to optimize for many objective functions, since its limited expressive capacity will make it more difficult for us to estimate and pursue the real objective – a limited set of images will effectively give us fewer points to estimate the PIR function from, and will reduce the space in which the model can easily produce images, thus reducing the possibility of getting very optimal images from the model. For example, a model which produces images of birds may not produce data points which provide good estimates of a PIR based on how much the image looks like a car, and even if it could, it may not be able to produce images which are "more car-like." If we are able to succeed in improving PIRs with this generative model, it is likely that a better generative model would yield even better results.

## 2.2 USER INTERACTIONS

We will show these images to users in a variety of ways, depending on our target domain. For the purposes of this paper, however, we will use simulated interaction data (see Section 3 for details). Of course, since showing images to users is an expensive prospect, we wanted to limit the size of the datasets we used to train the model. Typical datasets used to train vision models are on the order of millions of images, (e.g. ImageNet (Russakovsky et al., 2015)), but it is completely infeasible to collect user data on this number of images. We estimated that we could show 1000 images each 1000 times to generate our datasets. We used these dataset sizes and number of impressions for all experiments discussed here, and added noise to the PIRs that was binomially distributed according to the number of times each image was shown and the "true" PIR simulate from the objective.

## 2.3 PIR ESTIMATOR MODEL

The final component of our system is the PIR estimator model, which learns to predict PIR from a background image. We denote this model by $R : \text{image} \rightarrow [0, 1]$. We parameterize this model as a deep neural network. Specifically, we take the Inception v2 architecture (Szegedy et al., 2016), remove the output layer, and replace it with a fully-connected layer to PIR estimates. We initialize the Inception v2 parameters from a version of the model trained on [dataset redacted for blind review]. See Appendix C.3 for more details.

---

[1] In fact, we used an ACGAN (Odena et al., 2016) because it generated better initial images, but we present the results here in terms of a GAN for clarity. See Appendix C.1 for details of the ACGAN.

Why did we not make estimated PIR simply another auxiliary output from the discriminator, like class in the ACGAN (Appendix C.1)? Because the PIR estimator needs to be held constant in order to provide an accurate training objective. If the PIR estimates were produced by the discriminator, then as the discriminator changed to accurately discriminate the evolving generator images, the PIR estimates would tend to drift without a ground-truth to train them on. Separating the discriminator and the PIR estimator allows us to freeze the PIR estimator while still letting the discriminator adapt.

## 2.4 INTEGRATION

Once we have trained a PIR estimator model, we have to use it to improve the GAN. We do this as follows. Let $R$ denote the PIR estimator model, as above. Define $L_{\text{PIR}}$ to be the expectation of the estimated PIR produced over images sampled from the generator:

$$L_{\text{PIR}} = E_{z \sim \mathcal{N}(0,I), c \sim \mathcal{C}} \left[ R(G(z)) \right]$$

Then we simply supplement the generator loss by adding this term times a weight $w_{PIR}$, set as a hyperparameter:

$$L_{\text{Generator}} = L_{\text{fake image fools}} + w_{PIR} L_{PIR}$$

We set $w_{PIR} = 1000$ as this made the magnitude of the PIR loss and the other loss terms roughly comparable. Otherwise we used the same parameters as in the GAN training above, except that we reduced the learning rate to $10^{-6}$ to allow the system to adapt more smoothly to the multiple objectives, and we trained for 50,000 steps.

## 3 DATA

In this paper, we use simulated interaction data as proof of concept. This raises an issue: what functions should we use to simulate user interactions? Human behavior is complex, and if we already knew precisely what guided user interactions, there would be no need to actually collect human behavioral data at all. Since we don't know what features will guide human behavior, the next best thing we can do is to ensure that our system is able to alter the image generation model in a broad variety of ways, ranging from low level features (like making the images more colorful) to altering complex semantic features (such as including more plants in outdoor scenery). We also want to avoid hand-engineering tasks to the greatest extent possible. We present an overview of our approaches to simulating PIR data below, see Appendix C.4 for more details.

## 3.1 VGG FEATURES

The first approach we took to evaluating our system's ability to train for different features was to use activity from hidden layers of a computer vision model, specifically VGG 16 (Simonyan & Zisserman, 2014) trained on ImageNet (Russakovsky et al., 2015). In particular, we took the activity of a single filter in a layer of VGG relative to the overall activity of that layer. This approach to simulating PIRs has several benefits. First, it gives a wide variety of complex objectives that can nevertheless be easily computed to simulate data. Second, models like VGG exhibit hierarchical organization, where lower levels generally respond to lower-level features such as edges and colors, while higher levels respond to higher-level semantic features such as faces (Zeiler & Fergus, 2014), and the represented features relate to those in human and macaque visual cortex (Yamins et al., 2014). Thus VGG features give a wide range of objectives which we may relate to the human perception we wish to target.

There are some caveats to this approach, however. First, although the higher layers of CNNs are somewhat selective for "abstract" object categories, they are also fooled by adversarial images that humans would not be, and directly optimizing inputs for these high level features does not actually produce semantically meaningful images (Nguyen et al., 2014). Thus, even if our system succeeds in increasing activity in a targeted layer which is semantically selective, it will likely do so by adversarially exploiting particulars of VGG 16's parameterization of the classification problem (although the fact that we are not backpropagating through the true objective will make this harder). It is not necessarily a failure of the system if it exploits simple features of the objective it is given to increase PIRs – indeed, it should be seen as a success, as long as it is generalizes to novel images. However, success on this task does not necessarily guarantee success on modifying semantic content

when interacting with actual humans. It may be easier for the PIR estimator model (which is based on a CNN) to learn objectives which come from another CNN than more general possible objectives. The fact that adversarial examples can sometimes transfer between networks with different architectures (Liu et al., 2016) suggests that the computations being performed by these networks are somewhat architecture invariant. Thus CNN objectives may be easier for our estimator than human ones.

We have tried to minimize these problems to the greatest extent possible by using different network architectures (Inception V2 and VGG 16, respectively) trained on different datasets ([hidden] and ImageNet (Russakovsky et al., 2015), respectively) for the estimator and the objective. However, we cannot be certain that the network is not "cheating" in some way on the VGG 16 tasks, so our results must be considered with this qualification in mind. Despite this, we think that evaluating our system's ability to optimize for objectives generated from various layers of VGG will show its ability to optimize for a variety of complex objectives, and thus will serve as a useful indicator of its potential to improve PIRs from real users.

### 3.2 MULTIPLE FILTERS

After using our system on the tasks above, we noted that its performance was quite poor at layers 5, 6, and 7 of VGG compared to other tasks (see Fig. 2). This could suggest that our system was unable to capture the complex features represented at the higher levels of VGG. However, we also noticed that the feature representations at these layers tended to be quite sparse, so many of the simulated PIRs we generated were actually zero to within the bin width of our PIR estimator (see Appendix B Fig. 8 for a plot of how this affected learning). In order to evaluate whether the poor performance of our system at the higher layers of VGG was due to the number of zeros or to the complexity of the features, we created less sparse features from these layers by simply targeting a set of $k$ filters sampled without replacement from the layer, rather than a single filter. The single filter cases above can be thought of as a special case of this, where $k = 1$. To complement these, we also tried $k = 20$.

This can also be thought of as perhaps a more realistic simulation of human behavior, in the sense that it is highly unlikely that there is a single feature which influences human PIRs. Rather, there are probably many related features which influence PIR in various ways. Thus it is important to evaluate our system's ability to target these types of features as well.

### 3.3 COLORS

Finally, we also considered some simpler objectives based on targeting specific colors in the output images, or targeting vertical bands of two different colors, one in each half of the image, or three colors, one in each third of the image. These objectives provide a useful complement to the VGG objectives above. Although the single color objectives may be relevant to the classification task VGG 16 performs, the split color tasks are less likely to be relevant to classification. Note that it is important that we split the images along the width instead of the height dimension, as there may well be semantically relevant features corresponding to color divisions along the height dimension, e.g. a blue upper half and green lower half likely correlates with outdoor images, which would provide useful class information. By contrast, it is harder to imagine circumstances where different colors on the left and right halves of the image are semantically predictive, especially since flipping left to right is usually included in the data augmentation for computer vision systems. Thus success on optimizing for these objectives would increase our confidence in the generality of our system.

## 4 RESULTS

We present our results in terms of the change in mean PIR from 1000 images produced by the GAN before tuning to 1000 images produced after tuning, or in terms of the effect size of this change (Cohen's $d$, i.e. the change in mean PIR standardized by the standard deviation of the PIRs in the pre- and post-tuning image sets). We assess whether these changes are significant by performing a Welch's t-test (with a significance threshold of $\alpha = 0.001$) between the pre- and post-tuning PIRs.

Overall, our system was quite successful at improving PIRs across a range of simulated objective functions (see Fig. 2). Below, we discuss these results in more detail.

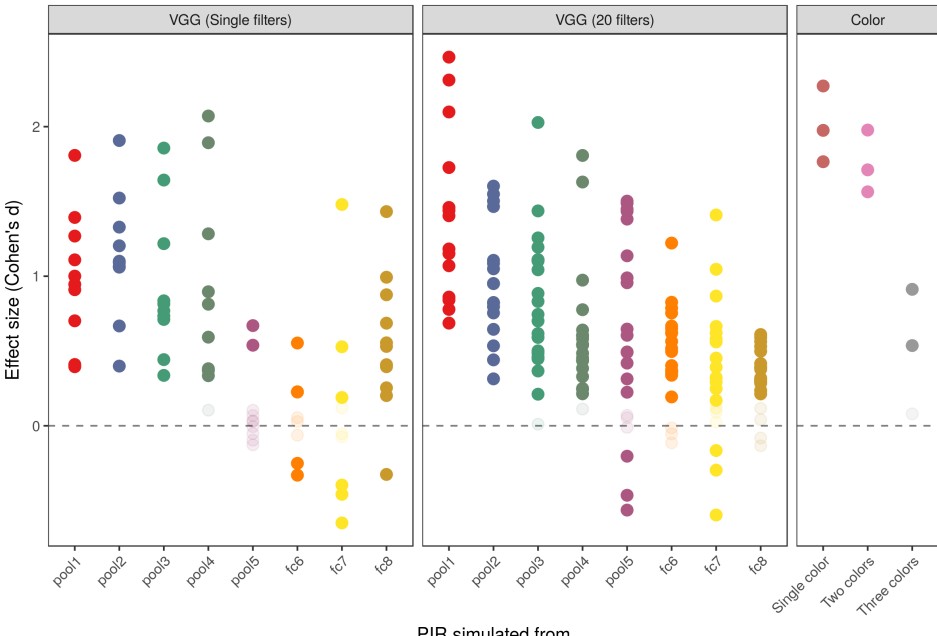

Figure 2: Effect size (number of standard deviations change in mean PIR) on a variety of tasks. Values greater than zero indicate improvement. Each panel represents a set of tasks (such as single filters from VGG), each color/column represents a subset (such as a single layer within VGG), and each point represents one objective (such as a single filter from that layer) for which we optimized the generative model. Points for which the change in mean PIR is not significant by a Welch's $t$-test with a threshold of $\alpha = 0.001$ are partially transparent.

## 4.1 VGG OBJECTIVES

Our system largely succeed at increasing PIRs on a variety of VGG objectives (see Fig. 2). However, there are several interesting patterns to note. First, the system is not particularly successful at targeting single filters from the pool5, fc6, and fc7 layers. However, we believe this is due to the fact that filters in these layers produce relatively sparse activation, see section 3.2. Indeed, the performance of the system seems much more consistent when it is optimizing for sets of 20 filters than for single filters.

Even when using 20 filters, however, there is a noticeable decline in the effect size of the improvement the system is able to make at higher layers of VGG ($\beta = -0.13$ per layer, $t = -7.8$, $p < 10^{-10}$ in a linear model controlling for initial standard deviation and percent zeros). This suggests that as the objectives grow more complex, the system may be finding less accurate approximations to them. However, the system's continuing (if diminished) success at the higher layers of VGG suggests that our model is capable of at least partially capturing complex objective functions.

## 4.2 COLOR OBJECTIVES

Overall, the system performed quite well at optimizing for the color objectives, particularly the single and two-color results (see Fig. 2). It had more difficulty optimizing for the three-color results, and indeed had produced only very small improvements after the usual 50,000 tuning steps for the generative model, but after 500,000 steps it was able to produce significant improvements for two out of the three objectives (these longer training results are the ones included here).

Because the color objectives are easiest to assess visually, we have included results for a variety of these objectives in Fig. 3. For the single color objectives, the improvement is quite clear, for example the images in Fig. 3d appear much more blue than the pre-training ones. For the two color objectives, it appears that the system found the "trick" of reducing the third color, for example the red-green split images in Fig. 3e appear much less blue than the pre-training images. Even on the three-color images where the system struggled, there are some visible signs of improvement, for example on the

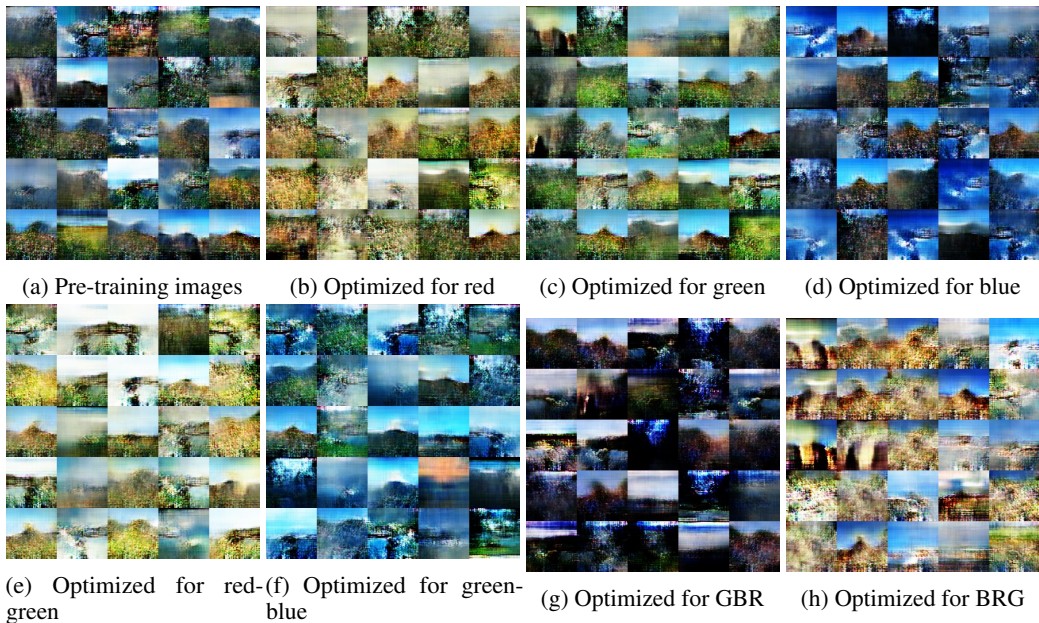

(a) Pre-training images    (b) Optimized for red    (c) Optimized for green    (d) Optimized for blue

(e) Optimized for red-green    (f) Optimized for green-blue    (g) Optimized for GBR    (h) Optimized for BRG

Figure 3: Color objective sample images. Samples are randomly drawn, not cherry-picked.

green-blue-red task the system has started producing a number of images with a blue streak in the middle.

### 4.3 SUPPLEMENTAL ANALYSES

We also conducted several supplemental analyses which can be found in detail Appendix A. In summary, the initial variability in the PIR of the images used to train the system is strongly correlated with the amount of improvement the system makes in the PIR, the system fairly consistently underestimates the performance it achieves (because of a detail of training procedure, see the Appendix), and iterating the process of improving PIRs yields better results for objectives from a lower layer of VGG but not a higher.

## 5 DISCUSSION

Overall, our system appears to be relatively successful. It can optimize a generative model to produce images which target a wide variety of objectives, ranging from low-level visual features such as colors and early features of VGG to features computed at the top layers of VGG. This success across a wide variety of objective functions allows us to be somewhat confident that our system will be able to achieve success in optimizing for real human interactions.

Furthermore, the system did not require an inordinate amount of training data. In fact, we were able to successfully estimate many different objective functions from only 1000 images, several orders of magnitude fewer than is typically used to train CNNs for vision tasks. Furthermore, these images came from a very biased and narrow distribution (samples from our generative model) which is reflective of neither the images that were used to pre-train the Inception model in the PIR estimator, nor the images the VGG model (which produced the simulated objectives) was trained on. Our success from this small amount of data suggests that not only will our system be able to optimize for real human interactions, it will be able to do so from a feasible number of training points.

These results are exciting – the model is able to approximate apparently complex objective functions from a small amount of data, even though this data comes from a very biased distribution that is unrelated to most the objectives in question. But what is really being learned? In the case of the color images, it's clear that the model is doing something close to correct. However, for the objectives derived from VGG we have no way to really assess whether the model is making the images better or

just more adversarial. For instance, when we are optimizing for the logit for "magpie," it's almost certainly the case that the result of this optimization will not look more like a magpie to a human, even if VGG does rate the images as more "magpie-like." On the other hand, this is not necessarily a failure of the system – it is accurately capturing the objective function it is given. What remains to be seen is whether it can capture how background images influence human behavior as well as it can capture the vagaries of deep vision architectures.

We believe there are many domains where a system similar to ours could be useful. We mentioned producing better webpage backgrounds and making more aesthetic images above, but there are many potential applications for improving GANs with a limited amount of human feedback. For example, a model could be trained to produce better music (e.g. song skip rates on streaming generated music could be treated as inverse PIRs).

## 5.1 TRADING IMAGE DIVERSITY FOR PIR

When tuning the GAN, the decrease in the PIR loss is usually accompanied by an increase in the generator loss, and often by a partial collapse of the generator output (for example, the optimized images generally seem to have fewer output modes than the pre-training images in Fig. 3). This is not especially surprising – because we weighted the PIR loss very highly, the model is rewarded for trading some image diversity for image optimality. Depending on the desired application, the weight on the PIR loss could be adjusted as necessary to trade off between producing images close to the data distribution and optimizing PIR. At its most extreme, one could down-weight the generator loss entirely, and train until the model just produces a single optimal image. However, the generator likely provides some regularization by constraining the images to be somewhat close to the real images, which will reduce overfitting to an imperfect estimate of the PIR function. Furthermore, in many settings we will want to generate a variety of images (e.g. backgrounds for different websites). For these reasons, we chose to keep the generator loss when tuning the GAN.

## 5.2 FUTURE DIRECTIONS

There are a number of future directions suggested by this work. A number of possible improvements are discussed in Appendix C.5. However, we also think this work has potential applications from the perspective of distillation or imitation approaches, which attempt to train one network to emulate another (Hinton et al., 2015; Parisotto et al., 2015, e.g), as well as from the perspective of understanding the computations that these vision architectures perform. As far as we are aware, these results are the first to show that a deep vision model can be tuned rapidly from relatively little data to produce outputs which accurately emulate the behavior of hidden layers of another deep vision architecture trained on a different dataset. This suggests both that the inductive biases shared among these architectures are causing them to find similar solutions (which is also supported by work on transferable adversarial examples (Liu et al., 2016, e.g.)), and that these networks final layers represent the computations of earlier hidden layers in a way that is somewhat accessible. It's possible that using our system with objectives from CNN layers as we did here might help to understand the features those layers are attending to, by analyzing the distribution of images that are produced. In this sense, our system can be thought of as offering a new approach to multifaceted feature visualization (Nguyen et al., 2016), because our system attempts to optimize a distribution of images for an objective and encourages diversity in the distribution produced, rather than just optimizing a single image.

## 6 CONCLUSIONS

We have described a system for efficiently tuning a generative image model according to a slow-to-evaluate objective function. We have demonstrated the success of this system at targeting a variety of objective functions simulated from different layers of a deep vision model, as well as from low-level visual features of the images, and have shown that it can do so from a small amount of data. We have quantified some of the features that affect its performance, including the variability of the training PIR data and the number of zeros it contains. Our system's success on a wide variety of objectives suggests that it will be able to improve real user interactions, or other objectives which are slow and expensive to evaluate. This may have many exciting applications, such as improving machine-generated images, music, or art.

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

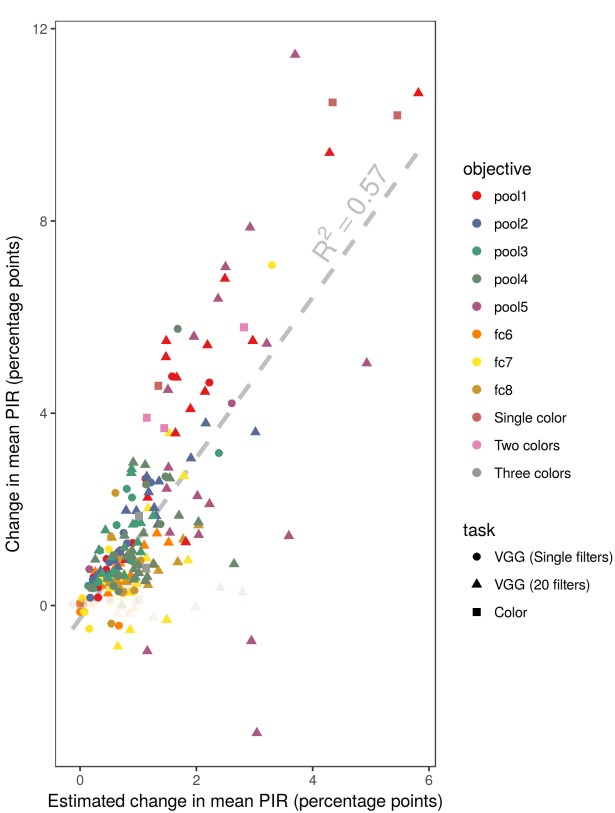

Figure 4: Change in mean PIR vs. estimated change in mean PIR. Points for which the change in mean PIR is not significant by a Welch's $t$-test with a threshold of $\alpha = 0.001$ are partially transparent.

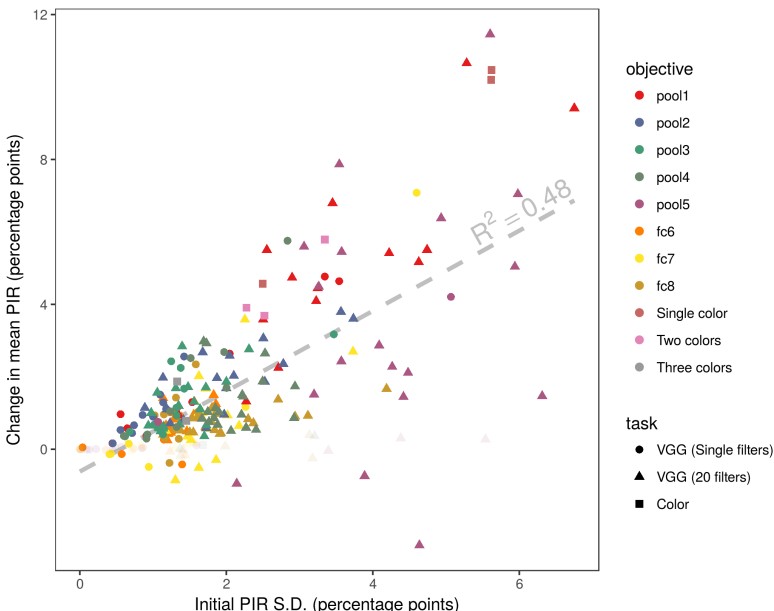

Figure 5: Change in mean PIR vs. initial variability. Points for which the change in mean PIR is not significant by a Welch's $t$-test with a threshold of $\alpha = 0.001$ are partially transparent.

## A  OTHER ANALYSES

### A.1  INTROSPECTION

Because $L_{PIR}$ is just the expected value of the PIR, by looking at $L_{PIR}$ before and after tuning the generative model, we can tell how well the system thinks it is doing, i.e. how much it estimates that it improved PIR. This comparison reveals the interesting pattern that the system is overly pessimistic about its performance. In fact, it tends to underestimate its performance by a factor of more than 1.5 ($\beta = 1.67$ when regressing change in mean PIR on predicted change in mean PIR, see Fig. 4). However, it does so fairly consistently. This effect appears to be driven by the system consistently underestimating the (absolute) PIRs, which is probably caused by our change in the softmax temperature between training the PIR estimator and tuning the generative model (which we empirically found improves performance, as noted above).

This is in contrast to the possible a priori expectation that the model would systematically over-estimate its performance, because it is overfitting to an imperfectly estimated objective function. Although decreasing the softmax temperature between training and using the PIR obscures this effect, we do see some evidence of this; the more complex objectives (which the system produced lower effect sizes on) seem to both have lower estimated changes in mean PIR **and** true changes in PIR which are even lower than the estimated ones (see Fig. 4). Thus although the system is somewhat aware of its reduced effectiveness with these objectives (as evidenced by the lower estimates of change in mean PIR), it is not reducing its estimates sufficiently to account for the true difficulty of the objectives (as evidenced by the fact that the true change in PIR is even lower than the estimates). However, the system was generally still able to obtain positive results on these objectives (see Fig. 2).

### A.2  INITIAL VARIABILITY

There is a general trend (see Fig. 5) that the variability in PIR in the initial dataset is strongly positively related with the change in PIR the system is able to produce ($\beta = 0.99$, $t = 10.6$, $p < 10^{-10}$, in a linear model controlling for initial mean PIR and initial percent of values that are zero). In fact, initial standard deviation explains about 50% of the variance in the change in mean PIR. This is perhaps not too surprising – more variability means that the generative model has capacity to produce higher PIR images without too much tweaking, and that the PIR estimator model gets a wider range of values to

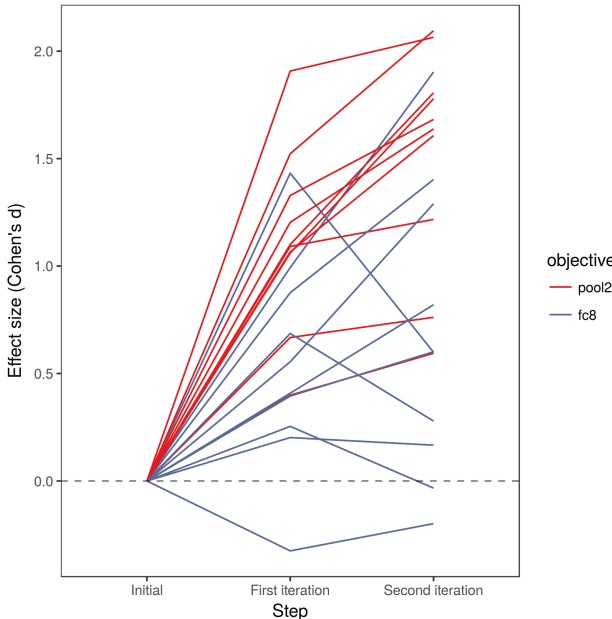

Figure 6: Effect size evolution over two iterations

learn from. Still, when attempting to use this system in practice, it is important to keep in mind that starting with a sufficiently expressive generative model will more likely produce better results than starting with a more limited model.

## A.3 ITERATION

Given that our model improves PIRs, an obvious question is whether we can iterate the process. Once we have increased PIRs, can we train a new PIR estimator on samples from our new generative model, and use that to increase PIRs again? If we could iterate this many times, we might be able to create much larger improvements in PIR than we can on a single step. On the other hand, it is possible that after a single step of optimization we will effectively have saturated the easily achievable improvement in the model, and further steps will not result in much improvement.

To evaluate this, we took the subset of models trained on single filters from VGG layers pool2 and fc8, and used the set of images generated from the post-tuning model along with the original set of images to tune their PIR estimators for another 25,000 steps, and then tuned the generative model using this updated PIR estimator for another 50,000 steps. We then evaluated them as before, see Fig. 6 for the results. The second iteration results were mixed, while the pool2 models all improved from the first step to the second, none of them improved as much as they had on the first step, and many of the fc8 models actually performed worse after the second step. However, it is possible that further hyperparameter tuning could improve these results, and it is certainly the case that running multiple steps of iteration and selecting the best model by experimentation could yield better results, so this is worth investigating further.

# B    SUPPLEMENTAL FIGURES

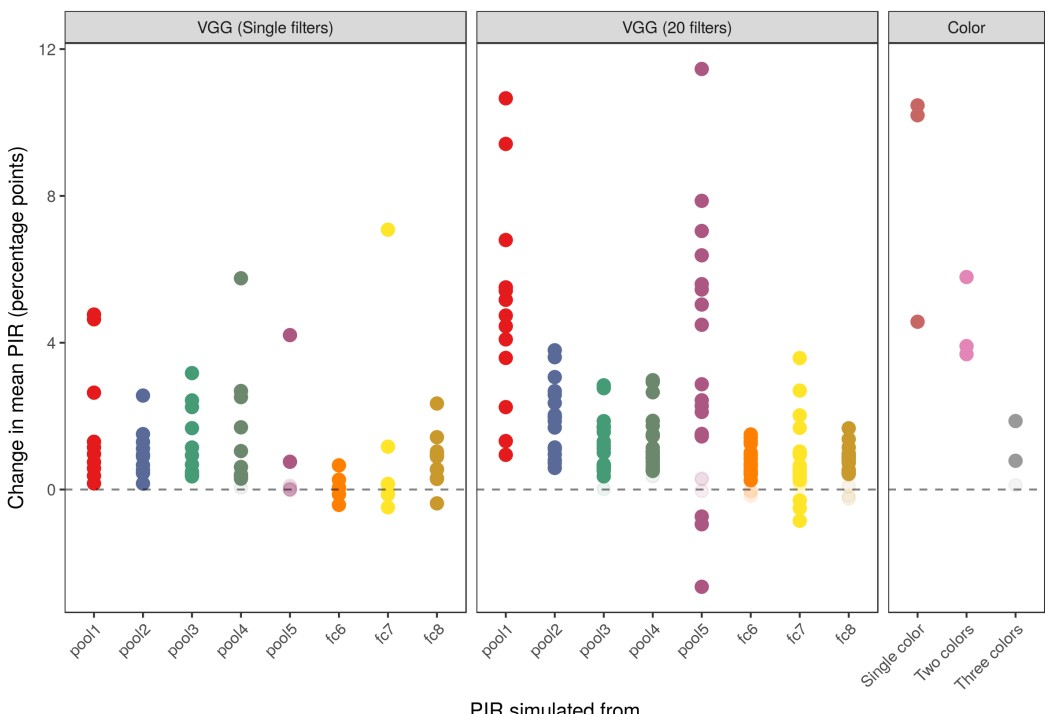

Figure 7: Change in mean PIR on a variety of tasks. Each panel represents a set of tasks (such as single filters from VGG), each color/column represents a subset (such as a single layer within VGG), and each point represents one objective (such as a single filter from that layer) for which we optimized the generative model. Points for which the change in mean PIR is not significant by a Welch's $t$-test with a threshold of $\alpha = 0.001$ are partially transparent.

# C    DETAILED METHODS

## C.1    ACGAN

We describe our results using the standard GAN framework for clarity, but we actually used an ACGAN (Odena et al., 2016), which allows for conditioning for various user-specific features. This requires the following adjustments. Defining $\mathcal{C}$ to be the set of possible classes and $\mathcal{I}_c$ to be the set of real images corresponding to a class $c \in \mathcal{C}$:

$$L_{\text{fake image}} = E_{z \sim \mathcal{N}(0,I), c \sim \mathcal{C}} \left[ \log P(D_{\text{source}}(G(z, c)) = \text{fake}) \right]$$

$$L_{\text{fake image fools}} = E_{z \sim \mathcal{N}(0,I), c \sim \mathcal{C}} \left[ \log P(D_{\text{source}}(G(z, c)) = \text{real}) \right]$$

$$L_{\text{real image}} = E_{c \sim \mathcal{C}, i \sim \mathcal{I}_c} \left[ \log P(D_{\text{source}}(i) = \text{real}) \right]$$

$$L_{\text{fake image class}} = E_{z \sim \mathcal{N}(0,I), c \sim \mathcal{C}} \left[ \log P(D_{\text{class}}(G(z, c)) = c) \right]$$

$$L_{\text{real image class}} = E_{c \sim \mathcal{C}, i \sim \mathcal{I}_c} \left[ \log P(D_{\text{class}}(i) = c) \right]$$

Then the discriminator and generator losses are modified as follows (letting $w_{fakeclass} = 1.5$ and $w_{realclass} = 1.0$):

$$L_{\text{discriminator}} = L_{\text{fake image}} + L_{\text{real image}} + w_{realclass} L_{\text{real class}}$$

$$L_{\text{Generator}} = L_{\text{fake image fools}} + w_{fakeclass} L_{\text{fake class}}$$

Note that unlike the standard ACGAN formulation given in (Odena et al., 2016), we do not include $L_{\text{fake class}}$ in the discriminator's loss, to keep the discriminator from cooperating with the generator on the classification task.

We modified the generator network by adding one-hot class inputs, and the discriminator by adding class outputs alongside the source output, as in (Odena et al., 2016).

## C.2 GENERATOR & DISCRIMINATOR

We parameterized the generator as a deep neural network, which begins with a fully-connected mapping from the latent (noise) space to a $4 \times 4 \times 512$ dimensional image, and then successively upsampled (a factor of 2 by nearest neighbor), padded and applied a convolution ($3 \times 3$ kernel, stride of 1) and a leaky ReLU ($\alpha = 0.2$) nonlinearity repeatedly. We repeated this process 5 times (except with no upsampling on the first step, and a tanh nonlinearity on the last), while stepping the image depth down as follows: 512, 512, 256, 128, 64, and finally 3 (RGB) for the output image. This means that the final output images were $64 \times 64$. We parameterized the discriminator as a convolutional network with 7 layers, 6 convolutions (kernels all $3 \times 3$; strides 2, 1, 2, 1, 2, 1; dropout after the 1st, 3rd, and 5th layers; filter depth 16, 32, 64, 128, 256, 512; batch normalization after each layer) and a fully connected layer to a single output for real/fake. We used a leaky ReLU ($\alpha = 0.2$) nonlinearity after each layer, except the final layer, where we used a tanh.

This GAN was trained on a dataset consisting of landscape images of mountains and coastlines obtained from the web. The generator was trained with the Adam optimizer (Kingma & Ba, 2015), and the discriminator with RMSProp. The learning rates for both were set to $10^{-5}$, and for Adam we set $\beta_1 = 0.5$. We used a latent size of 64 units. The model was trained for $1.1 \times 10^6$ gradient steps, when the generated images appeared to stop improving.

## C.3 PIR ESTIMATOR

Instead of predicting PIR as a scalar directly, we predict it by classifying into 100 bins via a softmax, which performs better empirically. This choice was motivated by noting that the scalar version was having trouble fitting some highly multi-modal distributions that appear in the data. We trained the PIR estimator with the Adam optimizer (learning rate $5 \cdot 10^{-4}$). When evaluating and when using this model for improving the GAN we froze the weights of the PIR estimator. We also reduced the output softmax's temperature to 0.01, so it was behaving almost like a max, which empirically improved results. Intuitively, a low softmax temperature in training allows the system to rapidly get gradients from many different output bins and adjust the distribution appropriately, whereas when actually using the system we want to be conservative with our estimates and not be too biased by low probability bins far from the modal estimate.

## C.4 SIMULATED DATA

### C.4.1 VGG FEATURES

The first approach we took to evaluating our system's ability to train for different features was to use activity from hidden layers of a computer vision model, specifically VGG 16 (Simonyan & Zisserman, 2014) trained on ImageNet (Russakovsky et al., 2015). In particular, we took the $\ell_2$ norm of the activity of one filter within a layer, and normalized it by the $\ell_2$ norm of the total layer's activity, i.e. letting $\text{VGG}_{l,f}(i)$ be the vector of unit activations in filter $f$ of layer $l$ of VGG 16 on image $i$, we computed PIR for that image and a given layer and filter $l^*, f^*$ as:

$$\text{PIR}_{l^*,f^*}(i) = \sqrt{\frac{|\text{VGG}_{l^*,f^*}(i)|_2^2}{\sum_{f \in l^*} |\text{VGG}_{l^*,f}(i)|_2^2}}$$

(Note that if we did not normalize by the activity in the whole layer, the system might be able to "cheat" to improve the PIR by just increasing the contrast of the images, which will likely increase overall network activity.) As noted above, we also added binomially distributed noise to these PIRs.

### C.4.2 MULTIPLE FILTERS

After using our system on the tasks above, we noted that its performance was quite poor at layers 5, 6, and 7 of VGG compared to other tasks (see Fig. 2). This could suggest that our system was unable to capture the complex features represented at the higher levels of VGG. However, we also noticed

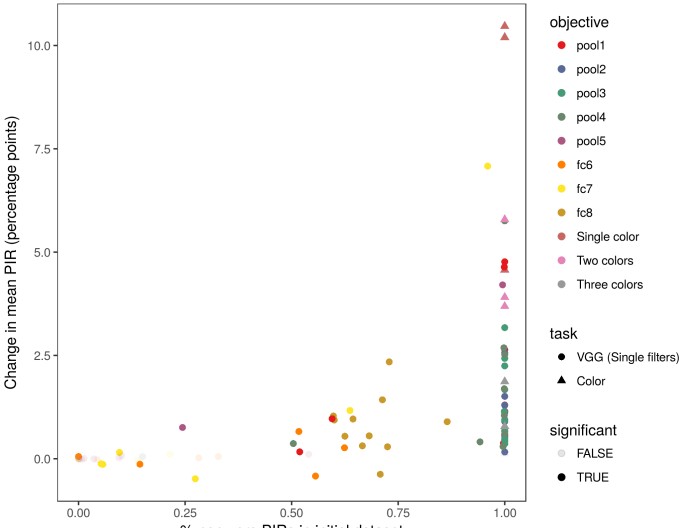

Figure 8: Percent non-zero PIRs vs. effect size (Single-filter VGG tasks and color tasks)

that the feature representations at these layers tended to be quite sparse, so many of the simulated PIRs we generated were actually zero to within the bin width of our PIR estimator. Respectively, these layers had only 20%, 25%, and 24% non-zero PIRs (collapsing across filters), and around half the filters in each (resp. 6, 4, and 5) were producing $> 90\%$ zero PIRs. (By contrast, the layer with the next greatest number of zero PIRs, fc8, still had 69% nonzero PIRs overall, and had no filters in which 90% or more of the PIRs were zero.) In a few cases on layers 5, 6, and 7, all of the generated PIRs were zero. This clearly makes learning infeasible, and indeed we noted that there was a strong relationship between number of non-zero simulated PIRs in the training dataset and the ability of our system to improve PIR (see Fig. 8). This is somewhat troubling, since probably most images in the real world will not produce a PIR that is truly zero.

in order to evaluate whether the poor performance of our system at the higher layers of VGG was due to the number of zeros or to the complexity of the features, we created less sparse features from these layers by simply targeting a set of $k$ filters sampled without replacement from the layer, rather than a single filter. We did this by taking the norm across the $k$ target filters, or equivalently by summing the squared norms of the $k$ filters before taking the square root, and then normalizing by the activity in the layer as before. Formally, letting $a_1, ..., a_k$ be a set of $k$ filter indices sampled without replacement from $\{0, ..., \text{number of filters in layer}\}$, we computed the PIR for an image as:

$$\text{PIR}_{l^*, f^*, k}(i) = \sqrt{\frac{\sum_{j=1}^{j=k} \left| \text{VGG}_{l^*, a_j}(i) \right|_2^2}{\sum_{f \in l^*} \left| \text{VGG}_{l^*, f}(i) \right|_2^2}}$$

The single filter cases above can be thought of as a special case of this, where $k = 1$. To complement these, we also tried $k = 20$. As above, we also added binomially distributed noise to these PIRs.

This can also be thought of as perhaps a more realistic simulation of human behavior, in the sense that it is highly unlikely that there is a single feature which influences human PIRs. Rather, there are probably many related features which influence PIR in various ways. Thus it is important to evaluate our system's ability to target these types of features as well.

### C.4.3   COLORS

Finally, we also considered some simpler objectives based on targeting specific colors in the output images. Analogously to the VGG features, we computed the PIRs from the vector norm of a given image in the targeted color, normalized by the total image value. We considered several objectives of this type:

**Single color:** Optimizing for a single color of output image, e.g., for red the objective would be.

$$\mathrm{PIR}_{\mathrm{red}}(i) = \sqrt{\frac{|i(:,:,\mathrm{red})|_2^2}{|\mathrm{i}(:,:,:)|_2^2}}$$

**Two color:** We split the image horizontally into a left and right half, and then computed PIR from one color in the left half and a different color in the right half.

$$\mathrm{PIR}_{\mathrm{red\ blue}}(i) = \sqrt{\frac{\left|i(:,:\frac{\mathrm{width}}{2},\mathrm{red})\right|_2^2 + \left|i(:,\frac{\mathrm{width}}{2}:,\mathrm{blue})\right|_2^2}{|\mathrm{i}(:,:,:)|_2^2}}$$

**Three color:** Similar to two color, but split the image into thirds, and computed PIR from a different color in each third.

(As above, we also added binomially distributed noise to these PIRs.) These objectives provide a useful complement to the VGG objectives discussed in section 3.1. Although the single color objectives may be relevant to the classification task VGG 16 performs, the split color tasks are less likely to be relevant to classification. Note that it is important that we split the images along the width instead of the height dimension, as there may well be semantically relevant features corresponding to color divisions along the height dimension, e.g. a blue upper half and green lower half likely correlates with outdoor images, which would provide useful class information. By contrast, it is harder to imagine circumstances where different colors on the left and right halves of the image are semantically predictive, especially since flipping left to right is usually included in the data augmentation for computer vision systems. Thus success on optimizing for these objectives would increase our confidence in the generality of our system.

## C.5 POSSIBLE IMPROVEMENTS

There are a number of techniques that could be explored to improve our system. As we mentioned above, iterating for multiple steps of PIR collection and generative model tuning is worth exploring further. Also, some form of data scaling might allow the system to perform better on tasks with low variance. We briefly tried normalizing all data for an objective to have mean 0.5 and standard deviation 0.25, but did not achieve particularly good results from this, possibly because there were many outliers getting clipped to 0 or 1. Still, there are many other possibilities for scaling data that could potentially result in some improvement in performance. Also, one alternative approach to training a GAN to produce high-PIR images would be to use the PIR estimator objective in the Plug & Play Generative Networks framework (Nguyen et al., 2017) instead of using it to tune the GAN. This could be an interesting direction to explore, but its success would probably depend on expressiveness of the initial generative model. With the mediocre model we started with, it's probably better to actually tune the model itself, which may allow it to explore parts of image space which it had not previously.

