# OpenReview forum: "Improving image generative models with human interactions"
_ICLR.cc/2018/Conference — Reject_

### Official Review · AnonReviewer1 · 2017-11-25
**interesting direction; but results are preliminary**

**Rating:** 4
**Confidence:** 5

**Review:**

+ Quality:
The paper discusses an interesting direction of incorporating humans in the training of a generative adversarial networks in the hope of improving generated samples. I personally find this exciting/refreshing and will be useful in the future of machine learning.

However, the paper shows only preliminary results in which the generator trained to maximize the PIR score (computed based on VGG features to simulate human aesthetics evaluation) indeed is able to do so. However, the paper lacks discussion / evidence of how hard it is to optimize for this VGG-based PIR score. In addition, if this was challenging to optimize, it'd be useful to include lessons for how the authors manage to train their model successfully.
In my opinion, this result is not too surprising given the existing power of deep learning to fit large datasets and generalize well to test sets.

Also, it is not clear whether the GAN samples indeed are improved qualitatively (with the incorporation of the PIR objective score maximization objective) vs. when there is no PIR objective. The paper also did not report sample quantitative measures e.g. Inception scores / MS-SSIM.

I'd be interested in how their proposed VGG-based PIR actually correlates with human evaluation.

+ Clarity:
- Yosinski et al. 2014 citation should be Nguyen et al. 2015 instead (wrong author order / year).
- In the abstract, the authors should emphasize that the PIR model used in this paper is based on VGG features.

+ Originality:
- The main direction of incorporating human feedback in the loop is original.

+ Significance:
- I think the paper contribution is lighter vs. ICLR standard. Results are preliminary.

Overall, I find this direction exciting and hope the authors would keep pushing in this direction! However, the current manuscript is not ready for publication.

---

> ### Author Response · Authors · 2018-01-02
> **Our results are exciting insofar as we demonstrate the ability to generalize complicated functions well from small datasets.**
>
> We agree that these results are preliminary, and we are presently working on testing this approach on real human data. Nevertheless we feel that these results are sufficient to show the promise of our method and provide a contribution to the literature.
>
> We think that the increase in PIR is somewhat surprising -- these are highly parameterized models which we are training with very few data points relative to standard deep network training paradigms (1000 images). That these highly parameterized models are able to fit a training set is not unexepcted. However, that the network is able to successfully generalize when targeting objectives based on single filters selected from the tens of thousands of filters and hundreds of millions of parameters in VGG, based on a dataset of only 1000 low-variability images was somewhat surprising and exciting to us.
>
> We would also like to clarify that the VGG based PIRs are used as a "ground-truth" for which to optimize, but our PIR estimator model treats the PIR function as a black box -- this is why it would be easy to substitute in human data for the VGG features. Thus the correlation of different VGG features with human evaluations is not particularly relevant.

---

### Official Review · AnonReviewer2 · 2017-11-28
**Review for "Improving image generative models with human interactions"**

**Rating:** 5
**Confidence:** 3

**Review:**

Summary:
This paper proposes an approach to generate images which are more aesthetically pleasing, considering the feedback of users via user interaction. However, instead of user interaction, it models it by a simulated measure of the quality of user interaction and then feeds it to a Gan architecture.

Pros:
+ The paper is well-written and has just a few typos: 2.1: “an Gan”.
+ The idea is very interesting.

Cons:

- Page 2- section 2- The reasoning that a deep-RL could not be more successful is not supported by any references and it is not convincing.

- Page 3- para 3 - mathematically the statement does not sound since the 2 expressions are exactly equivalent. The slight improvement may be achieved only by chance and be due to computational inefficiency, or changing a seed.

- Page 3- 2.2. Using a crowd-sourcing technique, developing a similarly small dataset (1000 images with 100 annotations) would normally cost less than 1k$.

- Page 3- 2.2.It is highly motivating to use users feedback in the loop but it is poorly explained how actually the user's' feedback is involved if it is involved at all.

- Page 4- sec 3 ".. it should be seen as a success"; the claim is not supported well.

- Page 4- sec 3.2- last paragraph.
This claim lacks scientific support, otherwise please cite proper references. The claim seems like a subjective understanding of conscious perception and unconscious perception of affective stimuli is totally disregarded.
The experimental setup is not convincing.

- Page 4. 3.3) "Note that.. outdoor images" this is implicitly adding the designers' bias to the results. The statement lacks scientific support.

- Page 4. 3.3) the importance of texture and shape is disregarded. “In the Eye of the Beholder: Employing Statistical Analysis and Eye Tracking for Analyzing Abstract Paintings, Yanulevskaya et al”
The architecture may lead in overfitting to users' feedback (being over-fit on the data with PIR measures)

- Page 6-Sec 4.2) " It had more difficulty optimizing for the three-color result" why? please discuss it.

- The expectation which is set in the abstract and the introduction of the paper is higher than the experiments shown in the Experimental setup.

---

> ### Author Response · Authors · 2018-01-02
> **Clarifications**
>
> Thank you for the comments, we have included some responses below, and added clarification to a few points in the paper.
>
> - Page 2- section 2- The reasoning that a deep-RL could not be more successful is not supported by any references and it is not convincing.
>
> We did not say that deep RL could not be more successful, we said this is not fundamentally a reinforcement learning problem, and is higher-dimensional than typical RL problems. There is no repeated temporal component to the interactions, which is what RL techniques are fundamentally based upon.
>
> - Page 3- para 3 - mathematically the statement does not sound since the 2 expressions are exactly equivalent. The slight improvement may be achieved only by chance and be due to computational inefficiency, or changing a seed.
>
> Mathematical equivalence does not imply computational equivalence -- For example there are many reasons (underflow, efficiency, etc.) that likelihood-based algorithms almost universally work with the log-likelihood rather than the direct likelihood, even though maximizing one is equivalent to maximizing the other. In our case, the curvature of the loss-function landscape could be quite dramatically changed, which could indeed result in the change in learning dynamics we observed.
>
> - Page 3- 2.2.It is highly motivating to use users feedback in the loop but it is poorly explained how actually the user's' feedback is involved if it is involved at all.
>
>  We hope that this is clarified immediately below, in the PIR estimator model section. The user data is precisely what the PIR estimator model is trained on.
>
> - Page 4- sec 3 ".. it should be seen as a success"; the claim is not supported well.o
>
> To clarify, we are pointing out that if the objective function we give our model has adversarial weaknesses, we should not be disappointed that it exploits them -- this may be the most efficient solution to the problem of increasing the PIR. It is a success to learn the objective function given, including its flaws. The remaining question is whether the model could learn objectives that do not have adversarial features.
>
> - Page 4- sec 3.2- last paragraph.
> This claim lacks scientific support [...] The claim seems like a subjective understanding of conscious perception and unconscious perception of affective stimuli is totally disregarded.
>
> It is complicated to untangle conscious and unconscious processes (see e.g. "Unconscious influences on decision making: A critical review", Newell & Shanks, Behavioral & Brain Sciences, 2014), and it is certainly beyond the scope of our paper to do so. However, since both are presumably supported by the visual cortex processes that are well modeled by the networks we are using as objectives (see Yamins et al., 2014, cited in our paper), we believe that our procedure is actually fairly well motivated as far as neuroscience's understanding of perception is concerned.
>
> - Page 4. 3.3) "Note that.. outdoor images" this is implicitly adding the designers' bias to the results. The statement lacks scientific support.
> - Page 4. 3.3) the importance of texture and shape is disregarded. The architecture may lead in overfitting to users' feedback (being over-fit on the data with PIR measures)
>
> We agree that the VGG tasks offer a more unbiased and complete set of features, that is why we ran them as well. We did not disregard these features. The color tasks just offer the opportunity to focus in on a specific feature and qualitatively evaluate performance visually on an intuitive and salient feature. The balance between fitting user feedback and fitting the original distribution can be shifted by changing the weights in the loss, as we mentioned in the discussion.
>
> - Page 6-Sec 4.2) " It had more difficulty optimizing for the three-color result" why?
>
> In the supplementary analyses, we show that the models improvement of the PIR is highly correlated with the initial variability in PIR (which is sensible, in general a function is better estimated by points that vary than points that are highly similar). The three color objectives simply have less initial variability in PIR, which makes it more difficult for the model to improve them. (This is also sensible -- there are not many natural images that appear with three different vertical color stripes, so of course there would be little variability in an objective which looks for three vertical color stripes.) In supplementary figure 5, you can see that the three color results are not outliers by any means when compared to the other objectives with similarly low initial variability.
>
> - The expectation which is set in the abstract and the introduction of the paper is higher than the experiments shown in the Experimental setup.
>
> We agree that these results are preliminary, nevertheless we feel that they are sufficient to show the promise of our method and provide a contribution to the literature.

---

### Official Review · AnonReviewer3 · 2017-12-01
**Practically relevant problem but paper is premature.**

**Rating:** 4
**Confidence:** 4

**Review:**

This paper proposes a technique to improve the output of GANs by maximising a separate score that aims to mimic human interactions.

Summary:
The goal of the technique to involve human interaction in generative processes is interesting. The proposed addition of a new loss function for this purpose is an obvious choice, not particularly involved. It is unclear to me whether the paper has value in its current form, that is without experimental results for the task it achieves. It feels to premature for publication.


More comments:
The main problem with this paper is that the proposed systems is designed for a human interaction setting but no such experiment is done or presented. The title is misleading, this may be the direction where the authors of the submission want to go, but the title  “.. with human interactions” is clearly misleading. “Model of human interactions” may be more appropriate.

The technical idea of this paper is to introduce a separate score in the GAN training process. This modifies the generator objective.  Besides “fooling” the discriminator, the generator objective is to maximise user interaction with the generated batch of images. This is an interesting objective but since no interactive experiments presented in this paper, the rest of the experiments hinges on the definition of “PIR” (positive interaction rate)using a model of human interaction. Instead of real interactions, the submission proposes to maximise the activations of hidden units in a separate neural network. By choosing the hierarchy level and type of filter the results of the GAN differ.

I could not appreciate the results in Figure 2 since I was missing the definition of PIR, how it is drawn in the training setup. Further I found it not surprising that the PIR changes when a highly parameterised model is trained for this task. The PIR value comes from a separate network not directly accessible during training time, nonetheless I would have been surprised to not see an increase. Please comment in the rebuttal and I would appreciate if the details of the synthetic PIR values on the training set could be explained.

- Technically it was a bit unclear to me how the objective is defined. There is a PIR per level and filter (as defined in C4) but in the setup the L_{PIR} was mentioned to be a scalar function, how are the values then summarized? There is a PIR per level and feature defined in C4.
- What does the PIR with the model in Section 3 stand for? Shouldn’t be something like “uniqueness”, that is how unique is an image in a batch of images be a better indicator? Besides, the intent of what possibly interesting PIR examples will be was unclear.
E.g., the statement at the end of 2.1 is unclear at that point in the document. How is the PIR drawn exactly? What does it represent? Is there a PIR per image? It becomes clear later, but I suggest to revisit this description in a new version.
- Also I suggest to move more details from Section C4 into the main text in Section 3. The high level description in Section 3.

---

> ### Author Response · Authors · 2018-01-02
> **Some clarifications of details of PIR**
>
> PIR:
>
> -  I found it not surprising that the PIR changes when a highly parameterised model is trained for this task. The PIR value comes from a separate network not directly accessible during training time, nonetheless I would have been surprised to not see an increase.
>
> We think that the increase in PIR is somewhat surprising -- these are highly parameterized models which we are training with very few data points relative to standard deep network training paradigms (1000 images). That these highly parameterized models are able to fit a training set is not unexepcted. However, that the network is able to successfully generalize when targeting objectives based on single filters selected from the tens of thousands of filters and hundreds of millions of parameters in VGG, based on a dataset of only 1000 images (sampled from a mediocre generative model) was somewhat surprising to us.
>
> - Technically it was a bit unclear to me how the objective is defined. There is a PIR per level and filter (as defined in C4) but in the setup the L_{PIR} was mentioned to be a scalar function, how are the values then summarized? There is a PIR per level and feature defined in C4.
>
> The PIR values defined in section C4 are the "ground-truth" values, that is, they represent the function(s) we are trying to get our PIR estimator to approximate. The l2 norms in the definitions make these ground-truth PIR values a scalar per-image for a given layer/filter choice. Each layer/filter choice represents a possible ground-truth function corresponding to a single point in the first panel of figure 2, we try many of these layer/filter ground-truths to evaluate the robustness of our approach.
>
> For a given layer/filter combination, we take the scalar ground-truth values per image and use them to draw noisy observations (simulating noisy data collection in the real world). We then use these to train the PIR estimator. We use this estimator as an additional loss, and backpropagate through it (with weights frozen) to improve the generator. Just as with the other losses in our experiment, we take the mean across the batch to reduce from a scalar value for each image (produced by the PIR estimator) to a single scalar loss. Hopefully this clarifies things.
>
> - What does the PIR with the model in Section 3 stand for? Shouldn’t be something like “uniqueness”, that is how unique is an image in a batch of images be a better indicator? Besides, the intent of what possibly interesting PIR examples will be was unclear.
> E.g., the statement at the end of 2.1 is unclear at that point in the document. How is the PIR drawn exactly? What does it represent? Is there a PIR per image? It becomes clear later, but I suggest to revisit this description in a new version.
>
> We wished to keep things as generic as possible, because our approach could be useful for many applications. Uniqueness might be one such possibility. Others you might consider are how aesthetically pleasing an image was (as assessed by human raters), how likely someone is to buy a monitor in a store when it displays a given image (as assessed by relative sale rates), how much users set a generated image as a background of their phone, etc. Going a little beyond the details of our model, this could also be used for services that stream generated music (skip rates are essentially a negative interaction rate), or any other services which generate content. Essentially any situation in which humans interact with the products of a generative model is a possible source of a "PIR" from our model's perspective. We hope these examples are helpful.
>
>
>
> Lack of human experiments:
>
> We agree that these results are preliminary, and we are presently working on testing this approach on real human data. Nevertheless we feel that these results are sufficient to show the promise of our method and provide a contribution to the literature.

---

### Decision · Program_Chairs · 2018-01-29
**ICLR 2018 Conference Acceptance Decision**

**Decision:**

Reject

**Comment:**

The reviewers agree that the idea of incorporating humans in the training of generative adversarial networks is interesting and worthwhile exploring. However, they felt that the paper fell short in providing strong support for their approach. The AC agrees. The authors are encouraged to strengthen their work and resubmit to a future venue.